# Exploring the Impact of COVID-19 Pandemic on Eating and Purchasing Behaviours of People Living in England

**DOI:** 10.3390/nu13051499

**Published:** 2021-04-29

**Authors:** Daniel A. Ogundijo, Ayten A. Tas, Bukola A. Onarinde

**Affiliations:** National Centre for Food Manufacturing, University of Lincoln, Holbeach PE12 7PT, UK; ATas@lincoln.ac.uk (A.A.T.); BOnarinde@lincoln.ac.uk (B.A.O.)

**Keywords:** COVID-19 pandemic, public health, sociodemographic variables, eating behaviour, purchasing habits

## Abstract

Consumers’ eating habits have changed significantly due to the anxiety and boredom from the reported cases and deaths of COVID-19, the change in work patterns, controlled food shopping, and the inability to meet loved ones during the lockdown. The magnitude of these changes in the eating behaviours and purchasing habits of consumers varies across different groups of people. This study provides empirical evidence of the effects of COVID-19 on the eating and purchasing behaviours of people living in England, which was assessed based on sociodemographic variables. A total of 911 participants were recruited by a market research company, while only 792 useable responses were included in this study. The participants, aged between 18 and 91 years, completed an online questionnaire, and the data were analysed using ordinal regression. Data were collected between October and December 2020. Male participants constituted 34.60%, females 63.89%, and others (other gender and those who prefer not to declare their gender) were 0.63%. The majority of participants’ ages fell into the ranges of 23–38 and 39–54. Participants aged 23 to 38 years had the greatest effect of COVID-19 on their purchasing decision of healthier foods, while participants in the age groups 55–73 and 74–91 were least affected. The amount of foods purchased during the pandemic decreased with increasing age. The amount of foods purchased by students, people in employment, and people from minority ethnic groups were greatly affected by the pandemic. All participants who stated that taking food supplements is not important during the pandemic were from the White ethnic group. The effects of the pandemic on purchasing healthier foods were greater in younger generations and participants in full- or part-time employment than participants who were retired and who were aged above 55. The participants with higher educational qualifications and those from minority ethnic groups were also more affected by the pandemic. We suggest further studies to monitor any changes in the effects of the ongoing COVID-19 pandemic on the eating and purchasing behaviours of consumers.

## 1. Introduction

People eat for different reasons, and the different foods they consume have an effect on their day-to-day activities. Eating behaviours vary from person to person and could be determined by array of physiological, sociological, psychological, or nutritional factors [1]. These needs were reported to be greatly affected by the effects of the pandemic of the coronavirus disease in 2019 (COVID-19). COVID-19 is caused by the severe acute respiratory syndrome coronavirus 2 (SARS-Cov-2), and this infectious disease was declared a pandemic by the World Health Organization [2]. The first case of COVID-19 was confirmed in the UK on 31 January 2020, and several measures (e.g., wash hands regularly, cover face by wearing a face covering in enclosed spaces, and make space by staying at least 2 m apart—or 1 m with a face covering or other precautions) were put in place to curb its spread. In the UK, the impacts of COVID-19 on people were reported as psychological imbalance [3], negative physical and mental wellbeing [4], and changes in weight and nutritional habits [5].

The global impacts of the COVID-19 pandemic could be seen in all areas of public health, such as the human, medical, biological, and social sciences, and has led consumers to have higher preference for organic foods, legumes, seafoods, fruits, and vegetables [6,7]. Consumers have been reported to now choose diets more to be safe and healthy than for any other reasons, such as environmental sustainability and improving personal appearance [8]. During the COVID-19 pandemic, Ammar et al. [9] saw a decrease in the consumption of alcoholic drinks, but eating patterns were out of control. It was also reported that Turkish university students claimed they ate a lot more during the pandemic, and that more attention was given to hygiene after food shopping [10].

Existing studies have pinpointed that anxiety, boredom, confinement to home during the lockdown, and regulated shopping by retailers contributed to a shift in eating behaviours during the COVID-19 pandemic [6,11,12,13]. Only a few of these studies measured the effects of COVID-19 on the sociodemographic characteristics of consumers. For empirical evidence on the impacts of COVID-19 on consumers’ health and diet during the pandemic, it was recommended that the levels of effects should be investigated in specific locations and among different categories of people [9,14]. This is because the ripple effects of COVID-19 on the economy, health, wellbeing, and diet differ from region to region and individual to individual. This study aims at measuring the impact of COVID-19 on the eating and purchasing behaviours of people living in England based on sociodemographic variables. The study appraises the empirical findings in line with the hypothesised opinions made on the impacts of the COVID-19 pandemic on consumers’ attitudes relative to age, ethnicity, education, gender, and the employment status of people that live in England. Peer-reviewed articles and grey literature with relevant theoretical frameworks that underpin food choice, consumer behaviours, and impacts of the COVID-19 pandemic were used to construct a conceptual framework that gives a general overview of the study [15,16,17,18].

## 2. Methods

An online survey deploying a validated structured questionnaire was used. The questionnaire was developed to collect information on the respondents’ diets and consumer behaviours during the COVID-19 pandemic. The recruitment of participants, questionnaire launching, and data gathering were carried out in November and December 2020.

The questions were developed to include the change in the amount of food products consumed, alterations in consumers’ purchasing decisions, and how the choice of foods and supplements were affected during the COVID-19 pandemic (Table 1). Two stages of validation (face and content validity) were used to test the reliability of the questions. Face validity was carried out to measure if the questions actually measured the objectives of the study, and the content validity assessed whether every question is representative of all the aspects of the construct. A validation sheet that comprises two sections was created. The first section is where the readability, comprehensiveness, and clarity of the questions were rated on a 1-to-5 Likert scale, ranging from highly favourable to highly unfavourable, and the second section is where the validation panel expressed their feelings on the questions and provided their comments and suggestions. The validation panel consisted of professionals in food and nutrition education, and public health. The set of data obtained from the validation study was analysed using the Statistical Package for Social Sciences (SPSS), version 27.

A pilot study was carried out to further test the validity and the reliability of the questionnaire, and 56 respondents took part in this pilot study. The results of the pilot study helped in adjusting the questions for more clarity and understanding. The participants’ responses were secured and were only viewed by the organisers of the survey. Prior to the launching of the survey, an ideal sample size of 750 was under consideration from the sample size calculation at a confidence level of 95% and margin error of 5%. Based on this scientific calculation, the result of this study is therefore believed to be a representation of the population of England.

The participants were recruited by a professional market research company, and only those individuals that reside in England were screened for participation. Participants below 18 years were excluded because they are a vulnerable population whose eating and purchasing decisions can easily be influenced [19,20]. The link to the survey was sent to the potential participants, and each participant gave their consent to participate before completing the questionnaire. Participants were assured that the data obtained from the study would only be used for statistical and academic purposes. The participants who did not meet the criteria of age or region, or who were not willing to consent, were automatically screened out by the software used. The incentive that was given to the participants was arranged by the market research company.

Before data analysis, the following hypotheses were made regarding the questions that the participants were asked about their eating and purchasing behaviours during the COVID-19 pandemic.

**Hypothesis** **1.**
*The amount of foods that consumers buy during the COVID-19 pandemic should be affected, because of the effect of stockpiling and restricted shopping by the supermarkets.*


**Hypothesis** **2.**
*The decision to buy healthier food products would be affected with increasing age.*


**Hypothesis** **3.**
*Food supplements and foods that support the immune system are expected to be consumed more by the elderly.*


**Hypothesis** **4.**
*The higher the education status, the less the decision on choice of healthier foods are affected by COVID-19.*


The Qualtrics software that was used to launch the questionnaire automatically analysed the data collected, and provided the mean, standard deviation, and variance of the data categories. Ordinal regression was used to determine the significance of the questions in relation to the demographic information of the participants. R software version R 3.6.2 was used to carry out ordinal regression on the survey data that was downloaded from Qualtrics. To avoid the likelihood of multiple comparisons of the responses, the *p*-value threshold was therefore reduced across the tests from the 5% that is normally used in statistical analysis. A false-discovery rate (FDR) analysis (as shown in Figure 1; the false discovery rate reduces “false positives” (wrongly significant results) to a manageable level when conducting “multiple comparisons”) was done on the data set to achieve the reduction, and an FDR cut off *p*-value of 0.0044 (*p* < 0.0044) was used instead of 0.05 to enhance the credibility of the results. The statistical significance was therefore set at a *p*-value of 0.0044 and all the coefficients from the ordinal regression below 0.0044 were significant. To further increase the accuracy of the results, the sociodemographic variables of the participants were used as covariates (a covariate is a possible predictive or explanatory variable of the dependent variable).

### Public Involvement

The perceptions of the members of the public on the research questions were provided during the validation, piloting, and main data collection stages. The responses of the participants were anonymous, and no participant could be identified by their responses. Nonetheless, in the participant information sheet, the participants were given the right to request the outcome of the study by contacting the researchers. The participants were given opportunities of refusal to participate and withdrawal from the survey before their responses were submitted. All the respondents gave consent to participate in the study, and all the information collected was for statistical and academic purposes only. There was no involvement of the public in the interpretation, reporting, or disseminating of the findings of this study.

## 3. Results

### 3.1. Sociodemographic Characteristics of the Participants

A total of 911 residents of England took part in the survey, which was greater than the ideal sample size, which was 750. As shown in Table 2, these participants comprised 63.89% (*n* = 506) females and 34.60% (*n* = 274) males, while 0.38% (*n* = 3) preferred not to declare their gender, and 0.25% (*n* = 2) said their gender could only be described in another way. The Millennial age classification that grouped participants into generations was used in this study, since people in the same generation are believed to have the same experience and behaviours [21]. The majority of the participants (43.94%) fell into the age group of 23–38 years, 31.57% were in 39–54, and 15.40% were in the age group of 55–73. The participants who were in the 18–22 years age group were 8.33%, while 0.63% were in the age group of 74–91, and 0.13% did not identify with any age group.

The UK government’s five categories of ethnicity3F (https://www.ethnicity-facts-figures.service.gov.uk/style-guide/ethnic-groups/ accessed on 12 January 2021) were used in this study, and most of the participants were from the White ethnic group (80.56%). Participants from the Asian or Asian British ethnic groups were 9.72%; Black, African, Black British, or Caribbean were 4.80%; and mixed or multiple ethnic groups were 2.53%. People who are categorised as “other ethnic group” in the UK, such as Arabs, were also represented, and made up 1.39% of the participants who took part in the study. Regarding their highest level of education, 97.48% of the participants had at least a secondary school qualification, 0.88% had some other qualification, and 1.64% preferred not to say the type of qualification they held. While 65.41% were in full- or part-time employment, 6.31% were retired, and 16.04% of were unemployed or currently seeking a job.

### 3.2. Eating and Purchasing Habits during the COVID-19 Pandemic

Table 1 clearly shows that there was a shift in consumers’ behaviours in making informed decisions on food choice, in the amount of food purchased, and in shopping during the COVID-19 pandemic. Only 25.63% of the participants said the frequency of their food shopping was not affected by the pandemic; however, the pandemic made 29.67% (*n* = 235) of the participants shop more online, while 29.42% still bought in-store but at a reduced frequency, and 15.28% shopped more in the stores. The pandemic was reported to have affected the choice of foods that 44.44% of participants eat or purchase, and the amounts of food products purchased was greatly affected in 15.53% (*n* = 123) of the participants. The decision to purchase healthier food products was affected in 77.77% of the participants, and only 12.50% of the participants said that the consumption of food supplements and foods that support the immune system was not at all important during the COVID-19 pandemic.

The ordinal regression analysis showed significant differences in the responses to some questions (Questions 4, 5, and 6) that were presented in Table 3, and Figure 1, Figure 2, Figure 3 and Figure 4. It could be seen that age, gender, education, ethnicity, and employment status had significant effects on consumers’ eating and purchasing habits. In fact, there were significant differences across the sociodemographic variables among the participants.

(a)Effects of COVID-19 pandemic on the amount of food purchased by participants (Question 4)

When participants were asked if the pandemic affected the amount of foods they purchase, significant differences were seen among different ages, ethnic groups, levels of education, and employment statuses. As shown in Figure 2, the effect of COVID-19 on the amount of foods bought during the pandemic decreased with increasing age groups. While 21% of the participants in the age group of 18–22 stated that the amount of foods they bought were greatly affected by COVID-19, all the participants whose ages were 74–91 claimed that they were only affected a little bit or moderately. Although 6% of the participants in the 55–73 group said they were greatly affected by the COVID-19 pandemic, 47% professed that they were not affected at all. The amounts of foods purchased during the COVID-19 pandemic were greatly affected in participants from the Black, Asian, and minority ethnic groups (BAME). Although Asians were the most affected, more than 25% of participants from each of the BAME groups were greatly affected, and about 50% of each said they were affected moderately or a little. The amounts of foods bought by the participants from White ethnic group were the least affected; in fact, 26% of them said that they were not affected at all.

The level of education of the participants also showed a significant variation on how the amount of foods bought during the pandemic was greatly affected. While the participants with the lowest level of education were the least affected, postgraduate qualification holders had the highest impact. A significant difference was seen on how the pandemic greatly affected the amount of foods bought by the secondary school qualification and college qualifications holders. Only 1% difference was seen between participants with college and first degree (undergraduate) qualifications. In addition, the participants who were students had the greatest effect of the COVID-19 pandemic on the amount of foods purchased compared to other employment statuses. The amount of the foods bought by the retirees during the pandemic were the least affected, and about 57% of them claimed that they were not affected at all.

(b)Effects of COVID-19 on participants’ attitudes to the consumption of food supplements (Question 5)

Although the degree of importance of taking food supplements and products that strengthen the immune system during the COVID-19 pandemic varied among people from the Black, mixed, and “other” ethnic groups, all the participants said that taking food supplements to support the immune system was important. The 12.55% of participants who said food supplements are “not important at all” were mostly from the White ethnic group (21%), and a few were from the Asian and Asian British ethnic group (1.8%). Apart from the participants that were from “other” ethnic group, such as Arabs, the greatest importance was given to the taking of food supplements by people from mixed or multiple ethnic groups, followed by people from the Black and Asian ethnic groups, as shown in Figure 3.

(c)Effects of COVID-19 on food purchasing decisions of the participants (Question 6)

As shown in Figure 4, the decision to purchase healthier foods was least affected by COVID-19 in participants within the oldest age groups (55–73 and 74–91). Participants whose ages were from 23 to 38 had the greatest effect of COVID-19 on their healthier foods purchasing decisions, followed by participants in the 18–22 group. The participants from the Asian, Asian British, or any Asian background (for example, Bangladeshi, Chinese, Indian, Pakistani) had the greatest effect of COVID-19 on food purchasing decisions, while those from White backgrounds were least affected.

All the participants who were not from Asian, Black, Mixed, or White ethnic groups had their healthier food purchasing decisions affected by the COVID-19 pandemic. Although people from mixed or multiple ethnic groups had the lowest number of participants who said COVID-19 greatly affected their decision on the choice of healthier foods, a considerable number of them had their decisions affected “moderately or a little bit”. Among the Black participants (Black African, Black British, or Caribbean—this includes any Black background), the percentage of people for whom COVID-19 affected their decision on healthier foods moderately and a little bit is lower than for Asian participants, but higher than for participants from the White ethnic group (this includes any White background).

In terms of employment status, COVID-19 had the greatest impact on students’ decision to buy healthier foods when compared with retirees and people in work. About 50% of the retirees who took part in the survey said that COVID-19 did not affect their decision to buy healthier foods, and they were generally least affected by COVID-19 in choosing healthier foods. Almost the same number of people in full-time and part-time employment said that the pandemic affected their food decisions greatly, but people in full-time employment whose decisions on food choice were moderately affected were more than those in part-time jobs. The majority of the participants who were looking for jobs (jobseekers) said that COVID-19 affected their purchasing decision “moderately or a little bit”.

## 4. Discussion

The current study demonstrated that eating and purchasing habits were influenced significantly during the COVID-19 pandemic, which can be observed clearly when responses are classified according to sociodemographic factors. These effects support the findings of Kaya [22] on how socio-economic and demographic variables affect consumers’ knowledge and attitudes. The effects of the pandemic on eating and purchasing habits are grouped as follows.

(a)Age

The outcome of the use of common sense by food retailers to give priority to elderly people during shopping hours during the COVID-19 pandemic can be inferred from the current study. Our data revealed that the amount of food and food products purchased by people who were in the 55–70 and 74–91 age groups was affected during the pandemic. Beyond the use of common sense, instead of rejecting the UK parliament’s petition [23] that would have given the elderly more privileges to access the supermarkets, our results suggest the need for government policy and frameworks on how the purchase of healthier foods by the elderly can be made easier in similar situations. The hypothesis that the decision to buy food supplements and food products that support the immune system would be greatly affected with increasing age could not be justified. This is because the significant difference observed in consumers’ attitudes to the consumption of food supplements and foods that support immunity was only related to ethnicity and did not vary significantly among the age groups. The participants aged 74–91 were least affected by COVID-19 in their decisions to buy healthier foods.

Nonetheless, the assertions of Johansen [24] and Oakes and Slotterback [25] that elderly consumers are more conscious of their health than young ones was reflected in the results. While elderly consumers prioritise the healthiness of their diets, younger consumers were said to focus on the effects of their foods on personal appearance and weight control [25].

(b)Education level

A study carried out with the residents of England reported that education level had an impact on eating behaviours, with a greater influence on people with lower qualifications [26]. It was therefore expected that people with higher qualifications would not be greatly affected by the COVID-19 pandemic in terms of making decisions on the amount and type of healthier foods purchased. Contrary to our opinion, the participants with the highest level of education in this study were the most affected. This may be because the participants with the highest qualifications are likely to be more health-conscious and obliged to buy more/a greater variety of healthier foods. The higher likelihood of people with higher educational qualifications to be gainfully employed and to earn a good income is also significant in making food choices. For example, the limited income of people with lower education has been linked with the consumption of energy-dense unhealthy diets, because those people usually cannot afford to purchase foods of good quality [27].

The effects of the COVID-19 pandemic on the amounts of foods purchased, consumption of food supplements, and decision to purchase healthier foods by students who were 18 years and above were not significant across the sociodemographic variables in this study. Nonetheless, other behavioural studies enumerated the sociological effects of COVID-19 on students across all the levels of education, ranging from pre-school to tertiary institutions in England [12,28]. The closure of schools that made children (except those who are vulnerable and whose parents are critical or key workers) (https://www.gov.uk/government/publications/coronavirus-covid-19-maintaining-educational-provision/guidance-for-schools-colleges-and-local-authorities-on-maintaining-educational-provision/ accessed on 3 February 2021) stay at home during the lockdown was reported to affect the eating routines of many school children.

(c)Employment status

The purchasing habits and decisions on choice of healthier foods were least affected in the older adults who were in the age groups 55–73 and 74–91, and those who were unemployed. This outcome is similar to that of the University of Michigan’s National Poll on Healthy Aging [29], where although in-store food purchases were found to reduce among older consumers who were between 50 to 80 years during the COVID-19 pandemic, no significant negative effects of the pandemic were recorded on their eating behaviours. The effects of the COVID-19 pandemic were observed differently on the nutrition and activities of older Dutch adults, where a substantial negative impact was seen on dietary habits [30].

Our survey could not justify why the eating and purchasing habits of the unemployed and older adult participants were least affected by the COVID-19 pandemic. This finding is contrary to the outcome of a qualitative study conducted in England, where all the participants whose eating behaviours were greatly affected before the COVID-19 pandemic were not in employment [31]. However, the social assistance supports received from the government could have helped them to maintain their dietary behaviours [32,33].

(d)Gender

The results did not show any statistical significance in food behaviours and purchasing habits between females, males, and those that described themselves in other ways. However, a higher number of females (63.89%, mostly from the White ethnic group) participated in the study. This could reflect the attitudes and behaviours of women and men toward food shopping and consumption, with a higher likelihood of women participating in food- and consumer-related surveys than men [34].

(e)Ethnic group

The reasons why COVID-19 greatly affected BAME participants in terms of decision-making and purchasing of healthier foods were not clear, but it may be a result of social, economic, or cultural factors [35]. The reported public health inequalities that are facing ethnic minorities also manifest in the risk and effects of the COVID-19 pandemic, and these health inequalities require urgent government attention and more empirical evidence-based studies [36]. Unlike studies in countries like Italy, Poland, Germany, Denmark, and Qatar [6,30,37,38] that measured the effect of the COVID-19 pandemic on the dietary behaviours among different ethnic groups, most available scientific findings on UK residents are epidemiological and clinical data [12,35]. In fact, empirical studies that adequately measure the effect of COVID-19 on eating and purchasing behaviours based on the sociodemographic characteristics of consumers in the UK are still limited. To the best of our knowledge, this is the first food behaviour study that critically explored the impacts of the COVID-19 pandemic on the socioeconomic variables of people that are resident in England.

We could assert from our data that the amount of healthier foods and supplements that consumers buy during the COVID-19 pandemic were greatly, moderately, or a little bit affected across all the sociodemographic variables. This evidence supports the guidance of the WHO that consumers should consume healthier foods during the pandemic [2]. Even though 22.22% of the participants claimed that their purchasing decisions were not affected by COVID-19, this was not the same for the remainder (77.78%) of the participants. This effect is also consistent with our hypothesis where we postulated an increase in the purchasing and consumption of healthier foods during the COVID-19 pandemic. The participants’ selection in this study was unbiased, randomised, and anonymous. The estimate of the sample size of the target population at a low FDR of 0.0044 and confidence level of 95% (which account for the deviation of the true proportion of the entire population) makes the generalisability of the findings of this study transferable to the entire population of England.

The main limitation of this study is that enough evidence could not be garnered and theoretically established on why the eating and purchasing behaviours were significantly affected during the COVID-19 pandemic when categorised into sociodemographic variables. For example, the impacts of the factors such as comorbidity and personal experience could not be measured through the study design; qualitative studies that will interact with the participants and provide scientific evidence to justify the effect of COVID-19 on their food behaviours are therefore recommended for future research. We suggest that separate studies should be done in other parts of the UK so that comparisons could be made with the data obtained from England, which is the most populated region.

## 5. Conclusions

The impacts of the COVID-19 pandemic are felt in many areas of people’s lives globally, and the way to prevent its spread and effects is to attract researchers’ pedagogical approaches. In this study, we have provided data on the impacts of the COVID-19 pandemic on food behaviours and consumers habits of people that live in England, stratified according to their age, gender, education level, ethnicity, and employment status. Although no scientific difference could be measured among the genders, the effects of the pandemic were more noticeable in younger generations, participants in full- or part-time employment, those with higher educational qualifications, and participants from minority ethnic groups. Because the COVID-19 pandemic is still ongoing, our data still need to be worked on in the future to know if the implications of the pandemic on consumers’ behaviours and habits are still the same in England, and perhaps in other regions of the UK.

## Figures and Tables

**Figure 1 nutrients-13-01499-f001:**
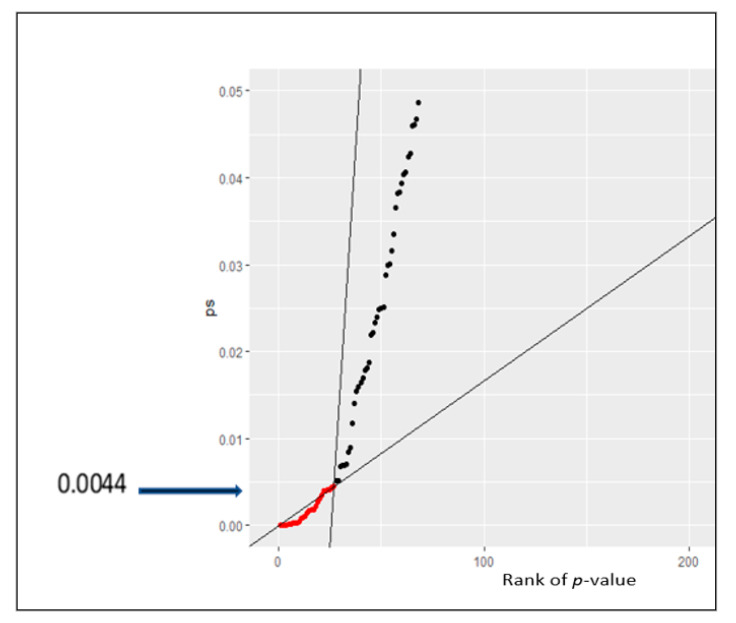
False discovery rate (FDR) for the validation tests.

**Figure 2 nutrients-13-01499-f002:**
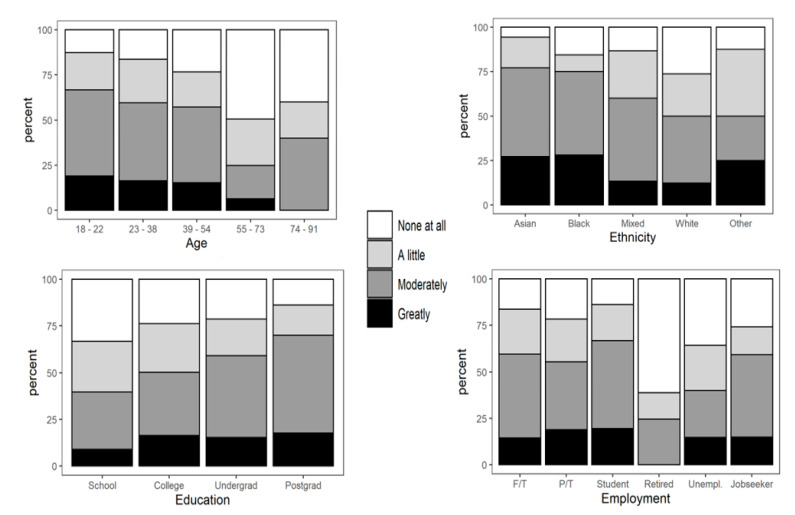
Impact of COVID-19 on the amount of foods purchased by participants grouped by age, ethnicity, level of education, and employment status (Question 4).

**Figure 3 nutrients-13-01499-f003:**
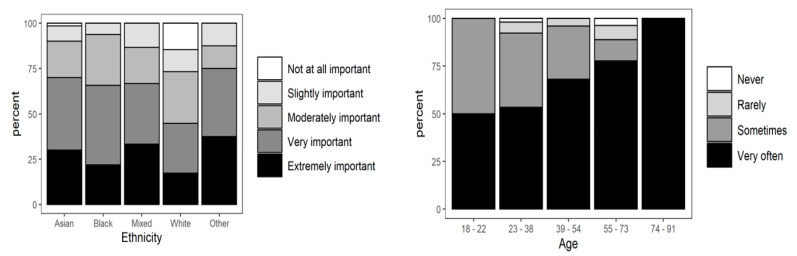
Effects of COVID-19 pandemic on the choice of food supplements and products that boost the immune system grouped by ethnicity and age (Question 5).

**Figure 4 nutrients-13-01499-f004:**
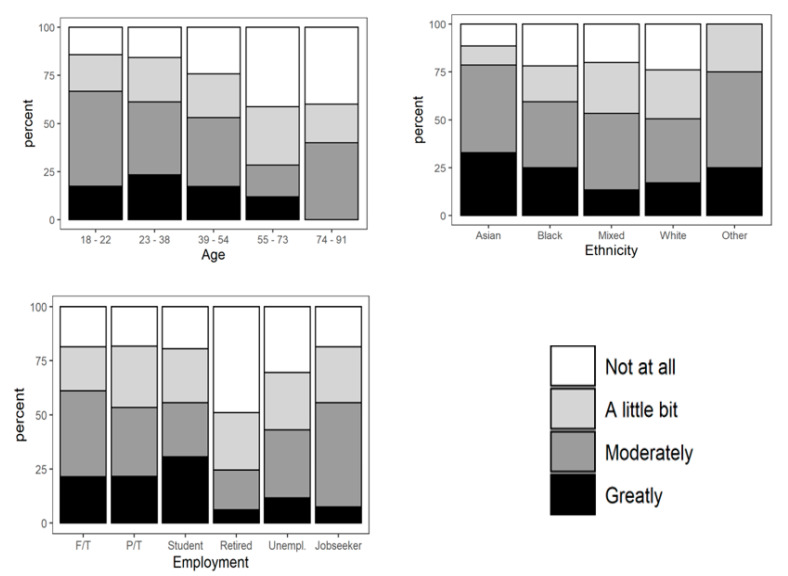
Effects of COVID-19 pandemic on purchasing decision of healthier foods grouped by age, ethnicity, and employment status (Question 6).

**Table 1 nutrients-13-01499-t001:** Participants responses (*n* = 792).

Questions	Responses, Presented as Percentage and Number of Participants				
1. How often did you shop for food before the Covid-19 pandemic?	Monthly	Fortnightly	Weekly	Every other day	Every day
	11.74% (*n* = 93)	14.77% (*n* = 117)	53.79% (*n* = 426)	17.05% (*n* = 135)	2.65% (*n* = 21)
2. How has Covid-19 pandemic affected the frequency of your food shopping?	I now shop more online	I now shop more in-store	I still buy in-store, but the frequency has reduced	No difference	
	29.67% (*n* = 235)	15.28% (*n* = 121)	29.42% (*n* = 233)	25.63% (*n*= 203)	
3. Has the Covid-19 pandemic affected the choice of the food you eat or purchase?	Yes		No		
	44.44% (*n* = 352)		55.56% (*n* = 440)		
4 *. To what extent did Covid-19 pandemic affect the amount of the food products that you buy?	Greatly	Moderately	A little bit	Not at all	
	15.53% (*n* = 123)	38.01% (*n* = 301)	23.23% (*n* = 184)	23.23% (*n* = 184)	
5 *. To what extent has the Covid-19 pandemic affected your decision on considering the purchase of healthier foods?	Greatly	Moderately	A little bit	Not at all	
	19.57% (*n* = 155)	33.96% (*n* = 269)	24.24% (*n* = 192)	22.22% (*n* = 176)	
6 *. How important do you think it is to take food supplements and products that boost the immune system during Covid-19 pandemic?	Extremely important	Very important	Moderately important	Slightly important	Not at all important
	20.33% (*n* = 161)	29.67% (*n* = 235)	26.39% (*n* = 209)	11.11% (*n* = 88)	12.50% (*n* = 99)

* Questions with responses that were significant when categorised based on sociodemographic variables.

**Table 2 nutrients-13-01499-t002:** Sociodemographic characteristics of the participants (*n* = 792).

Variable	Characteristics	Frequency	Percent (%)
Age	18–22	66	8.33
	23–38	348	43.94
	39–54	250	31.57
	55–73	122	15.40
	74–91	5	0.63
	Prefer not to say	1	0.13
Gender	Male	274	34.60
	Female	506	63.89
	In another way	2	0.25
	Prefer not to say	3	0.38
Ethnicity	Asian or Asian British	77	9.72
	Black, African, Black British or Caribbean	38	4.80
	Mixed or multiple ethnic groups	20	2.53
	White. This includes any White background	638	80.56
	Another ethnic group, for example, Arab	8	1.01
	Prefer not to say	11	1.39
Highest level of education	Secondary school	179	22.60
	College or vocational training	249	31.44
	Undergraduate	210	26.52
	Postgraduate	134	16.92
	Other	7	0.88
	Prefer not to say	13	1.64
Employment status	Employed/self-employed full time	365	46.09
	Employed/self-employed part-time	153	19.32
	Full time student	39	4.92
	Retired	50	6.31
	Unemployed	99	12.50
	Currently looking for work	28	3.54
	Other	41	5.18
	Prefer not to say	17	2.15

**Table 3 nutrients-13-01499-t003:** Significant *p* and false discovery rate (FDR) cut-off values for Questions 4 (“To what extent did the COVID-19 pandemic affect the amount of the food products that you buy?”), 5 (“How important do you think it is to take food supplements and products that boost the immune system during the COVID-19 pandemic?”), and 6 (“To what extent has the COVID-19 pandemic affected your decision on considering the purchase of healthier foods?”), with significant coefficients.

Question	Age	Education	Gender		Ethnicity				Employment				
			Male vs. Female	Other vs. Female	Black. vs. Asian	Mixed vs. Asian	White vs. Asian	Others vs. Asian	Retired vs. Full Time	Unemployed vs. Full Time	Part Time vs. Full Time	Jobseeker vs. Full Time	Student vs. Full Time
4	**0.00014** (**−0.36224**) *****	0.00182 (**0.21880**) *****	0.87537 (0.05930)	0.87537 (−0.37601)	0.00033 (−0.33370)	0.00033 (−0.77946)	0.00033 (**−1.02145**) *****	0.00033 (−0.57295)	0.00226 (**−1.07286**) *****	0.00226 (**−0.69365**) *****	0.00226 (−0.07882)	0.00226 (−0.37938)	0.00226 (−0.28791)
5	0.08377 (−0.15917)	0.00510 (**0.19110**) *****	0.36787 (−0.10886)	0.36787 (−1.39311)	0.00019 (−0.27200)	0.00019 (0.07469)	0.00019 (**−0.88832**) *****	0.00019 (0.26095)	0.04602 (**−0.90428**) *****	0.04602 (−0.24688)	0.04602 (−0.30120)	0.04602 (−0.23764)	0.04602 (−0.49209)
6	0.08235 (**−0.36309**) *****	0.29583 (0.07191)	0.19271 (−0.16172)	0.19271 (1.80691)	0.00175 (**−0.85021**) *****	0.00175 (−0.90174)	0.00175 (**−0.92459**) *****	0.00175 (0.06229)	0.00309 (**−0.81888**) *****	0.00309 (**−0.74525**) *****	0.00309 (−0.19288)	0.00309 (−0.55293)	0.00309 (−0.51965)

The values are presented as a(b), where a is *p* < 0.05, and b is the FDR cut-off value. Bolded coefficients with * are more than twice the standard error and significant at the FDR cut-off level of 0.0044, which is a reasonable criterion for judging the significance of the coefficients.

## Data Availability

Please contact the corresponding author for this information.

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
