# Peer review of "Exploring the Impact of COVID-19 Pandemic on Eating and Purchasing Behaviours of People Living in England"

_nutrients, 2021, doi:10.3390/nu13051499_

Round 1
Reviewer 1 Report
Interesting work.The study provides evidences on the effects of Covid-19 on the eating and purchasing behaviours of people living in England. Online questionnaire is well described as well as methods and results section. Interesting, the effects of the pandemic on purchasing healthier foods differed by age group and employment status . Also, regression analysis showed how sociodemographic variables such as gender, education, and ethnicity had significant effects on consumers`eatingand purchasing habits. Tables and figures are well presented. Conclusions are clear and, correctly, highlight the limitations of the study.
Author Response
Dear Reviewer,
No comments provided. Thank you for your feedback. We appreciate your positive comments on the methods, analysis, conclusion, and the general structure of the manuscript.
Kind regards,
All authors

Reviewer 2 Report
The study is very small to representative but the way it is demonstrated is very nice it can be good to go back to the people and ask now how purcheses changes now
Author Response
Dear Reviewer,
Please find our responses to the comments, which are indicated in red.
Kind regards,
----------------------------------------------------------------------------------------
Reviewer’s comment:
The study is very small to representative but the way it is demonstrated is very nice it can be good to go back to the people and ask now how purchases changes now. The sample size calculation (power calculation) was done at the beginning of the study, and the number of participants used in the study is more than the ideal sample size. We therefore believe that the number of the participants used is a representative of the population of England. This clarification can be seen from Lines 98 -102.
Your suggestion to go back to the participants and investigate the current state of their purchasing behaviour is appreciated and we shall think about it in the future.

Reviewer 3 Report
In this study the Authors aim at measuring the impact of Covid-19 on the eating and purchasing behaviours of a sample of English inhabitants, taking into account the socio-demographic variables relative to age, ethnicity, education, gender, and employment status.
The Authors clearly show that during the Covid-19 pandemic consumers’ behaviours underwent significant changes in making decision on food choice, in the amount of food purchased and shopping habits, as clearly evidenced by the significant differences in the responses to questions 4, 5 and 6, in relation to age, gender, education, ethnicity, and employment status of the participants.
1. Major critical issues:
- although the stratification of the data demonstrates the effect produced by the pandemic on consumer habits, nevertheless the Authors should clarify whether these percentages were attributable to the pandemic per se, rather than to the specific variables considered.
- Furthermore, the Authors should specify whether the percentages found regarding age, ethnicity, education, gender and employment status faithfully reflect those attributable to the general population of England, rather than to the sample recruited in the study.
- Finally, the Authors merely focused their attention on the socio-demographic aspects underlying food behaviors and consumers habits, without considering the economic impact of these variables on the purchasing capacity of the participants.
2. Minor critical issues:
- page 3, line 92: "Responses, presented as Percentage and (number of participants), the use of parentheses is not clear;
- the layout of Table 3 does not facilitate the comparative reading of the reported data. It would be advisable to re-edit it more conveniently;
- the panels indicated in figures 2, 3 and 4 could be better grouped to facilitate reading in relation to their legends and textual comment.
Author Response
Dear Reviewer,
Please find our responses to your valuable comments, which are indicated in red.
Kind regards,
-------------------------------------------------------------------------------------------
Reviewer 3
Comments and Suggestions for Authors
In this study the Authors aim at measuring the impact of Covid-19 on the eating and purchasing behaviours of a sample of English inhabitants, taking into account the socio-demographic variables relative to age, ethnicity, education, gender, and employment status.
The Authors clearly show that during the Covid-19 pandemic consumers’ behaviours underwent significant changes in making decision on food choice, in the amount of food purchased and shopping habits, as clearly evidenced by the significant differences in the responses to questions 4, 5 and 6, in relation to age, gender, education, ethnicity, and employment status of the participants.
Major critical issues:
- Although the stratification of the data demonstrates the effect produced by the pandemic on consumer habits, nevertheless the Authors should clarify whether these percentages were attributable to the pandemic per se, rather than to the specific variables considered. All the findings were attributable to the pandemic, and this is reflected in the topic and abstract, and has been clearly showcased all through the manuscript. For example, it was categorically stated in Line 2 that the study explored the impacts of Covid-19 pandemic on the behaviours of the participant. Our recommendation in Lines 27 - 28 also suggests that more studies on the impacts of Covid-19 on consumers’ behaviours should be done.
- Furthermore, the Authors should specify whether the percentages found regarding age, ethnicity, education, gender and employment status faithfully reflect those attributable to the general population of England, rather than to the sample recruited in the study. The sample size calculation (power calculation) was done at the beginning of the study, and the number of participants used in the study is more than the ideal sample size. We therefore believe that the number of the participants used is a representative of the entire England population. This clarification can be seen from Line 98 -102. To the best of our knowledge, this is the first food behaviour study that critically explored the impacts of Covid-19 pandemic on the socioeconomic variables of people that are resident in England (Lines 368-376). We therefore could not compare our data with any other study that is specific to England except for the behaviours of consumers before the pandemic (Lines 320-322, 353-357)
- Finally, the Authors merely focused their attention on the socio-demographic aspects underlying food behaviours and consumers habits, without considering the economic impact of these variables on the purchasing capacity of the participants. The higher income of people with higher education qualifications and gainful employment has been included in the manuscript (Lines 327-331). We recognised the economic impact of the variables on the participants, but enough evidence could not be established because of the research instrument that was used. More empirical information would have been obtained if qualitative methods (i.e., interviews or focus groups) were used, and this is one of the limitations, as outlined in Lines 393-401.
Minor critical issues:
- Page 3, line 92: "Responses, presented as Percentage and (number of participants), the use of parentheses is not clear. The title of Table 1 (Line 93) and the description of responses statement have been reworded as: Table 1. Participants` responses (n=792) and Responses, presented as percentage and number of participants, respectively.
- The layout of Table 3 does not facilitate the comparative reading of the reported data. It would be advisable to re-edit it more conveniently. The Table slightly drifted during the formatting and the values appeared not aligned, this has now been adjusted. The focus is on the bolded values which were significant, however other values which were not significant were presented so as to showcase the differences during the discussion.
- The panels indicated in figures 2, 3 and 4 could be better grouped to facilitate reading in relation to their legends and textual comment. Figures 2, 3, and 4 have been regrouped and their explanations updated. The legends have been adjusted to avoid repetition.
